# Foaming of Polycaprolactone and Its Impregnation with Quercetin Using Supercritical CO_2_

**DOI:** 10.3390/polym11091390

**Published:** 2019-08-23

**Authors:** Ignacio García-Casas, Antonio Montes, Diego Valor, Clara Pereyra, Enrique J. Martínez de la Ossa

**Affiliations:** Department of Chemical Engineering and Food Technology, Faculty of Science, University of Cádiz, International Excellence Agrifood Campus (CeiA3), 11510 Puerto Real (Cádiz), Spain

**Keywords:** supercritical CO_2_, polycaprolactone, quercetin, foaming, impregnation, release profile

## Abstract

Foamed polycaprolactone impregnated with quercetin was carried out with a batch foaming technique using supercritical CO_2_. The experimental design was developed to study the influence of pressure (15–30 MPa), temperature (308–333 K), and depressurization rate (0.1–20) on the foam structure, melting temperature, and release tests of composites. The characterization of the experiments was carried out using scanning electron microscopy, X-ray diffractometer, and differential scanning calorimetry techniques. It was observed that the porosity created in the polymer had a heterogeneous structure, as well as the impregnation of the quercetin during the process. On the other hand, controlled release tests showed a significant delay in the release of quercetin compared to commercial quercetin.

## 1. Introduction

The use of CO_2_ as a blowing agent in foaming has become increasingly popular in the scientific community to encapsulate, fabricate scaffolds, and produce delivery-controlled systems. This is due to the fact of CO_2_’s properties as non-toxic, inexpensive, and reusable, as well as its high dissolution in polymers [1].

The supercritical foaming process has certain advantages; for example, the use of CO_2_ over other blowing agents, such as chlorofluorocarbon (CSF) or the non-use of co-solvent, make this technique an environmentally friendly process. This process exploits the compressibility power of CO_2_ on the polymer to create a porous structure in the polymer. Initially, the polymer is saturated with a gas at constant conditions of pressure and temperature. Once saturated, the system is driven (i.e., quenched) to a supersaturated state, usually decreasing the pressure, although if the process has been taken to low temperatures, the temperature could be increased. This causes nucleation and growth of the porous cells within the polymeric matrix [2]. There are different aspects to consider when a polymer is under supercritical conditions, such as the swelling, the melting point variation, and foaming [3,4,5]. Particularly in the case of semi-crystalline polymers, when the CO_2_ molecules are dissolved, they facilitate the mobility of the chains. This favors the reorganization of the polymer chains, lowering the crystallization temperature and improving crystallinity [6]. In the case of the effect caused by swelling in the matrix, this leads the polymer chains to reorganize into a more extended configuration. All this causes a greater facility in the formation of crystalline structures and, therefore, it usually decreases the melting temperature [7].

Polycaprolactone (PCL) is a semi-crystalline polyester with a melting point (*T*_m_) of 329–334 K and a glass transition temperature (*T*_g_) of 213 K [8]. Many studies have investigated the effect of pressure, temperature, time, and, to a lesser extent, the depressurization rate on the foaming and impregnation processes [3,9,10,11] of PCL with supercritical CO_2_ (scCO_2_). Several studies have reported that a polymer could be impregnated by an active substance by a swelling process when scCO_2_ is vented. Since the lower diffusivity of the active substance regarding to the scCO_2_, it can be trapped by the polymeric matrix [3,12]. This fact provokes the idea that the foaming and impregnation processes happen during a one-step process. Growth factors were impregnated into hybrid PCL-starch scaffolds to study its delivery [13]: dexamethasone was impregnated into PCL/silica nanoparticles [14] and into silk fibroin aerogel/PCL for bone regeneration [15], nimesulid was impregnated into PCL via a one-step foaming/impregnation process [16], strontium was impregnated into PCL scaffolds [17], and TiO_2_ was impregnated into PCL for removal textile dyes [18]. Even Tsivintzelis et al. [19] reported the use of co-solvent (ethanol) in the formation of more homogenous scaffolds with PCL.

Quercetin (*Q*) is a flavonoid present in many fruits and vegetables [20]. This flavonoid is highlighted for its antioxidant action, but it has different benefits such as anti-inflammatory, antibacterial, cardiovascular health, and anticancer effects [21,22]. This antioxidant compound has been used by several authors with scCO_2_ in addition to different supercritical techniques, such as a supercritical antisolvent process to co-precipitate or encapsulate the quercetin with polymer [23,24] and a supercritical impregnation process (SSI) to impregnate the quercetin into different polymers or a porous matrix [25,26]. The SSI processes substitutes the liquid organic solvent with a supercritical fluid giving it the advantage that the final product is completely free of any residual solvent contamination. Thus, controlled quercetin release could be prepared as a safe process.

This paper focused on the use of a biodegradable polymer PCL as a coating agent for a delivery system to control the release of an active substance. In this article, a one-step foaming/impregnation process with PCL and quercetin was conducted. Moreover, the effect of pressure, temperature, and depressurization rate on the foaming process, the melting temperature and melting heat of the composites, and on the release profiles of quercetin were evaluated.

## 2. Materials and Methods 

### 2.1. Materials

Polycaprolactone (pellets, average *M*_n_ 45.000 g mol^−1^) was provided by Sigma–Aldrich (Madrid, Spain). Quercetin (C_15_H_10_O_7_) was purchased from Sigma–Aldrich (Madrid, Spain). CO_2_ with a minimum purity of 99.8% was supplied by Linde (Barcelona, Spain). 

### 2.2. Experimental Design 

A design of experiment (DOE) was carried out in order to identify the main factors that should be taken into account in the foamed PCL–quercetin preparation using supercritical CO_2_. A full factorial design 2^3^ was performed. The complete design consisted of 11 experimental points that included 8 factor points and three replications at the center point (experiments 1, 2 and 8). The main factors were selected with adequate ranges for this design. The responses of the design were melting temperature (*T*_m_), mg of released quercetin with regard to g of PCL, and melting heat (*H*_m_). The design was conducted with Modde 5.0 software.

Pressure (*P*), temperature (*T*), and depressurization rate (*D*_r_) were identified as the main factors that can directly influence the foaming process of the features of the formed composites and, thus, on quercetin release profiles. The two levels for each factor are shown in Table 1. The levels of pressure and temperature were selected to evaluate the influence of different supercritical CO_2_ densities on the achievement of different degrees of plasticization of the polymer in including quercetin into its structure. Depressurization rate levels were selected according to the minimum and maximum set points available from the equipment supplier. The contact times were fixed to 1 h in order to ensure that there was enough time to form a polymer plasticization and to keep the costs reasonable.

### 2.3. Foaming and Impregnation with scCO_2_

The experimental setup of the apparatus used to foam and impregnate is represented in Figure 1. The plant consisted of a CO_2_ bottle, a high-pressure pump to boost the CO_2_, a thermal bath to cool the CO_2_ to its liquid state, a heat exchanger to fix the temperature, a stainless-steel cell with a volume of 257 mL where the foaming and impregnation was carried out. Initially, each sample was formed by 500 mg of PCL and 10 mg of quercetin mixed physically into an aluminum foil support (ratio 50:1 PCL:*Q*). Then, it was introduced into a stainless-steel cell, after which CO_2_ was pumped into the cell to reach the optimum conditions for the foaming process at the same time that the temperature was adjusted. This study was carried out in the range of 15–30 MPa of pressure, temperatures of 308–333 K, and a foaming/impregnation time of 1 h. Once the process time had finished, the output valve was opened to vent the CO_2_ in a depressurization range of 0.1–20 MPa min^−1^.

### 2.4. Sample Characterization 

#### 2.4.1. Scanning Electron Microscopy

Foamed/impregnated polymer samples were processed with a FEI TENEO scanning electron microscope (SEM) in order to analyze morphology. The images were obtained on a cross-section of the sample on a carbon tape and then coated with a 12 nm film of gold prior to analysis. 

#### 2.4.2. X-Ray Diffraction

X-ray diffraction (XRD) analysis was performed on a Bruker D8 Advance diffractometer in order to determine the amorphous or crystalline nature of the foamed polymer loaded with quercetin after the supercritical foaming/impregnation process. All diffraction patterns were scanned from 10° to 50° in 2 *θ* angles with a step size of 0.02° and 1 s as the step time. CoK_a_ was the used source.

#### 2.4.3. Differential Scanning Calorimetry

Thermal property measurements were taken with a DSC Q100 (TA Instruments). Samples were heated to 373 K at a heating rate of 10 K min^−1^, then samples were cooled at 10 K min^−1^, and finally heated to 373 K at the same rate. All tests were carried out under the protection of a nitrogen (99.999%) atmosphere and following the standards for DSC, ISO 11357-3 [2018].

### 2.5. In Vitro Release Test

Simulated fluid (SF) was produced in the laboratory, which was a solution of 6.8 g/L of monobasic potassium phosphate in distilled water and adjusted to pH 6.8 ± 0.1 with 0.2 N NaOH.

An amount of 0.25 g of the polymer impregnated with quercetin was dissolved in 40 mL of SF in order to carry out the release test. The release tests were carried out at a temperature of 310 K and a stirring speed of 165 rpm, and 3 mL aliquots were withdrawn and filtered at time intervals of 5, 15, 30, 60 min, and then every hour until 480 min. Finally, an extra aliquot was withdrawn after 24 h. The released quercetin was measured with a Shimadzu UV-VIS mini spectrophotometer at λ = 365. The calculation was based on that reported by Zhu et al. [27] to calculate correctly the concentration of released quercetin.

## 3. Results and Discussion

### 3.1. Analysis of the Design of Experiments

The values for the coefficient of determination *R*^2^ (goodness of fit), *Q*^2^ (goodness of prediction), degree of freedom (*DF*), sum of squares (*SS*), and mean square (*MS*) are shown in Table 2. According to *R*^2^, the model was well adjusted and the goodness of prediction (*Q*^2^) was good for *T*_m_ and mg of released *Q*. However, *H*_m_ was badly adjusted, thus, no conclusions can be established from the model for this response. Analyzing the values of *F* and *p* (a high *F*-value and a low *p*-value less than 0.05), it can be ensured that the model was significant for the two first responses [28], but not significant for *H*_m_. It can be seen from Table 2 that, statistically, the model’s lack of fit was due to the probability of a lack of fit being non-significant at 95%. The experimental design and the responses of the experiments are provided in Table 3.

The effects on responses had positive and negative signs, as can be seen in Table 1. The sign of each effect indicates whether the response increased or decreased when the experiments changed from a low to a high level. For instance, an increase in temperature from low to high led to a considerable decrease in *T*_m_. Depressurization rate had the second most marked effect on *T*_m_ but, in this case, the trend was the opposite: an increase in *D*_r_ led to an increase in *T*_m_. With regard to the released quercetin, the most marked effect was observed for temperature followed by *D*_r_ and pressure but, in this case, the effect’s difference was lower. These effects had different signs, which indicate that released quercetin increased with pressure and temperature and decreased with *D*_r_.

Taking into account the calculated effects in Table 1, *H*_m_ decreased when pressure and temperature decreased and increased when *D*_r_ decreased. However, according to Table 2, no significant difference in *H*_m_ was found and it was not possible to establish any trends.

Coefficients of the linear model for the significant responses, *T*_m_ and released quercetin given in Table 4 show that temperature and *D*_r_ were the main significant variables for both responses (*p* < 0.05).

### 3.2. Foaming and Impregnation Experiments

All experiments showed visually an inhomogeneous impregnation along the structure of the PLC, as can be seen in Figure 2. The bottom of the polymers seemed to accumulate the highest quantity of *Q*. The low solubility of *Q* in scCO_2_ [29] may have contributed to the higher amounts of *Q* found at the bottoms. Experiments 5 and 10, conducted at the maximum temperature (333 K) and at the lowest depressurization rate (0.1 MPa) but with different pressures (15 and 30 MPa, respectively), showed the same external texture, presenting the highest foaming effect. Furthermore, experiments 3, 4, 7, and 11 exhibited the lowest foaming effect. These experiments were carried out at the minimum level of temperature (308 K) revealing the importance of temperature in the foaming process. Finally, experiments 6 and 9, conducted at the maximum temperature (333 K) and the fastest depressurization rate (0.1 MPa), exhibited a similar aspects to experiments 1, 2, and 8 (central point experiments at 22.5 MPa, 320.5 K and 2.5 MPa min^−1^) with foaming, a surface formed by small bubbles and a lower volume than experiments 5 and 10.

### 3.3. Characterization

#### 3.3.1. Scanning Electron Microscope (SEM)

The experiments conducted were screened with SEM (Figure 3). Raw PCL was also analyzed in order to compare possible changes in the porosity structure. Unlike raw PCL, the images of the composites show the formation of unstructured porous structures due to the adsorption of scCO_2_ into the semi-crystalline polymer, and the subsequent heterogeneous nucleation at the interface between the crystalline and amorphous regions and the homogeneous nucleation in the amorphous phase [30]. A structural difference depending on the depressurization rate was observed. Experiments 5, 7, 10, and 11 (Figure 3), carried out at a low depressurization rate (0.1 MPa min^−1^), present a surface with big pores (around 100 µm) and lower cell number densities. However, the experiments 3, 4, 6, and 9 (Figure 3), conducted at a fast depressurization rate (20 MPa min^−1^), showed a high presence of micro and nano pores with higher cell number densities. Given the impact of pressure and temperature, experiments 3 and 4 had the higher density of pores, with an abundance of pores on the surface area, as can be seen in Figure 3. This was expected because experiments 3 and 4 were carried out at a lower temperature than experiments 6 and 9 (from 333 to 308 K). Particular attention should be paid to experiment 3, carried out at a higher pressure (30 MPa) and at the same temperature (308 K) and depressurization rate (20 Mpa min^−1^), where a higher cell density can be observed. It is known [3,19,31] that the solubility of CO_2_ in the polymer increases when the temperature decreases and pressure increases; this causes an increase in the amount of CO_2_ in the PCL, causing higher cell density and smaller cell diameter. Also, the nucleation theory [2] explains that, at higher temperatures and lower pressures, the energy barrier to nucleation increases; consequently, the generation of nuclei is rougher, causing the reduction in the final cell density.

#### 3.3.2. X-ray Diffraction

The XRD analysis (Figure 4 and Appendix A) shows all experiments with a typical XRD diffraction pattern of raw PCL, and the same pattern for the treated PCL and foamed/impregnated samples. The PCL is a semi-crystalline polymer with peaks of high intensity at 21.25° and 23.7°. The supercritical foaming process increases the intensity of the crystallinity peak and decreases the presence of quercetin, reducing its crystallinity. This occurs because the presence of quercetin blocks the formation of large crystallites [14,32]. The peak of raw *Q* showed a wide quantity of characteristic peaks situated at 2*θ* = 12.3°, 12.9°, 17.75°, 21.95°, 24.5°, 26.5°, and 27.4° [22,33]. Comparing the PCL foamed/impregnated diffraction patterns with the quercetin diffraction pattern (Figure 4), it is observed that the peaks of raw *Q* are not presented. It could be due to the fact that *Q* was changed to an amorphous nature, although this seems to be unlikely. Thus, it was probably caused due to the inhomogeneous impregnation of quercetin or the quite higher ratio polymer: quercetin.

#### 3.3.3. Differential Scanning Calorimetry

Figure 5 represents the DSC analyses of the samples, which are more different. In every plot, a simple peak corresponding to the melting temperature is shown. Moreover, melting heat (*H*_m_) was proportioned by DSC. Both *T*_m_ and *H*_m_ are collected in Table 3. In general, it must be taken into account that the pressurization of CO_2_ including quercetin leads to a polymer with a *T*_m_, in some cases, with eight degrees of difference. With regard to *H*_m_, it was not possible to establish parameter trends due to the statistical parameter collected in the ANOVA table, which pointed to quite bad adjustment and prediction (Table 3). Besides, the highest *H*_m_ was found when the highest values of pressure, temperature, and *D*_r_ were set (Experiment 9). In Figure 6, the main effect plots on *T*_m_ can be seen. An increase in temperature led to a composite with notably lower *T*_m_, and an increase in pressure had the same trend but with lower influence. This can be explained due to the fact that when higher operating conditions are set, CO_2_ is able to penetrate into the polymer chain where *Q* can be located, leading to a less stable structure and, thus, lower *T*_m_. Moreover, the soaking of CO_2_ into the polymer structure improves the crystallinity and then decreases *T*_m_ [6,7]. However, the depressurization rate had the opposite trend, so when *D*_r_ increased, composites with higher *T*_m_ formed. In the depressurization step, nucleation of the bubbles and diffusion of the CO_2_ through the polymer competed. At a rapid depressurization rate, more nuclei were formed and the growth and coalescence of pores were minimized before vitrification led to a more uniform pore size distribution [34,35]. This pore structure was related to the more thermal stability (higher *T*_m_) of composites according to the obtained results.

The nature of interaction among variables was also explored, as can be seen in Figure 7 where interaction plots are shown. In general, no interaction was observed between pressure and temperature and pressure and *D*_r_. It seems that only *D*_r_ and temperature had a mild interaction, so at a low level of temperature, *D*_r_ had a significant influence on *T*_m_, increasing when *D*_r_ was augmented. However, at a high temperature, the *D*_r_ effect was neglected.

Two main effects plots are shown in Figure 8. The surface slope was made by two main effects and the possible twist by their two-factor interaction. The twist was observed as expected when *D*_r_ and temperature were plotted. In general, lower pressure and lower temperature together with higher *D*_r_ lead to composites with higher *T*_m_.

### 3.4. In Vitro Release Test

The release of *Q* from polymer was studied for all experiment (Figure 9a) and compared with raw quercetin (Figure 9b). In general, the release of *Q* from polymer was slower than pure dissolution of quercetin alone. Release of *Q* from polymer depends on the formed composites and, thus, on pressure, temperature, and *D*_r_. For instance, experiment 3 and 6 differed almost two times in mg of released *Q*. It is necessary to investigate the main effects, interactions, and response surface plots to shed light on the parameter influence on release profiles.

In Figure 10, the main effect plots on released quercetin can be seen. An increase in pressure leads to a composite, where the mg of the released quercetin is increased. Temperature had the same trend due to the increase in cell diameter and lower cell density at a higher temperature process. In the same way, at higher conditions, CO_2_ penetrates more easily into the polymer structure increasing the space between polymer chains where quercetin can be located. In this sense, part of the quercetin would be more accessible and not entrapped in the rigid polymer structure to release into the media. However, *D*_r_ had the opposite trend, so an increase in *D*_r_ led to a lower amount of released quercetin. It was expected due to the fact that the samples obtained with higher *D*_r_ presented high densities of porosity in their surface areas and the sizes of the pores were smaller than in the other experiments.

All this supports that experiment 5, with the higher pressure, temperature, and low depressurization rate (30 MPa, 333 K, and 0.1 MPa min^−1^), was the sample with the highest released *Q* after 8 and 24 h. Figure 9b shows the release of raw *Q* in comparison with experiment 1, processes under the supercritical condition of 22.5 MPa, 320.5 K, and 2.5 MPa min^−1^. The release profile shows a significant difference; while the raw *Q* needed 1 h to dissolve, the *Q* impregnated in the foamed PCL needed 8 h to release and dissolve the same quantity of *Q*; this means it was eight times slower. This demonstrates the utility of PCL to coat an active principle, in this case *Q*, to control the release process for in vitro tests in simulated fluids with a pH around 7.

The nature of interaction among factors was also explored, as can be seen in Figure 11. In general, no interaction was observed between *D*_r_ and pressure and temperature. However, a mild interaction was observed between pressure and temperature, so at low pressure, the temperature effect was almost neglected on the released quercetin. This fact could be observed in the two main effects plots (Figure 12). It can be seen that higher pressure and temperature and lower *D*_r_ are recommended to release the highest amount of quercetin from the composites.

## 4. Conclusions

The foaming and impregnation one-step process of PCL and quercetin was successfully conducted. Pressure, temperature, and depressurization rate were evaluated for the foaming, melting temperature, melting heat, and the amount of released quercetin. Melting heat data had no significant differences and no trends were established. A high temperature is recommended to obtain a pronounced effect of foaming. Pressure and depressurization rate have an important influence on responses. The generated PCL/quercetin composites with higher pore density and smaller size were achieved for higher pressure and depressurization rate and lower temperatures (300 bar, 308 K, and 20 MPa min^−1^). This study also showed that experiments carried out at lower pressure and temperature together with a higher depressurization rate lead to a higher melting temperature. An increase in pressure and temperature lead to a composite which releases higher amounts of quercetin. However, the depressurization rate had the opposite trend; thus, an increase in the depressurization rate led to lower amounts of released quercetin. Release profiles showed that quercetin takes five times longer to dissolve the same amount of quercetin in the first 8 h, demonstrating the efficacy of using PCL to control quercetin release and its possible use with other medical or pharmaceutical compounds.

## Figures and Tables

**Figure 1 polymers-11-01390-f001:**
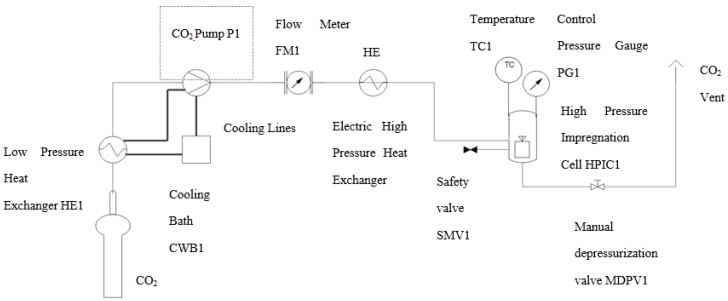
Schematic foaming/impregnation plant.

**Figure 2 polymers-11-01390-f002:**
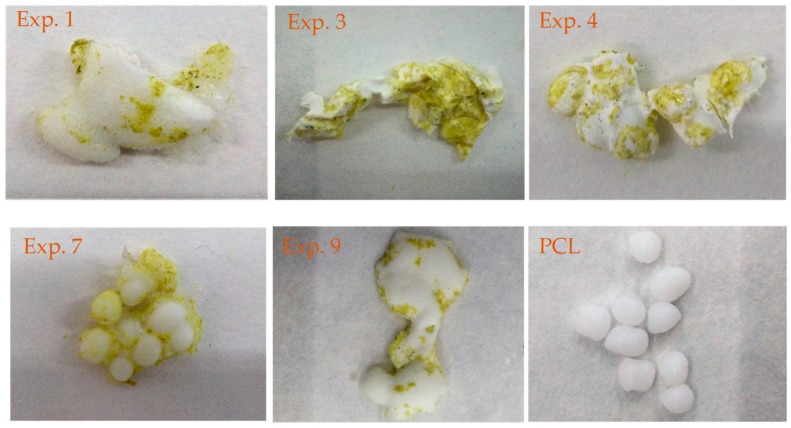
Photos of experiments 1, 3, 4, 7, and 9 and raw PCL.

**Figure 3 polymers-11-01390-f003:**
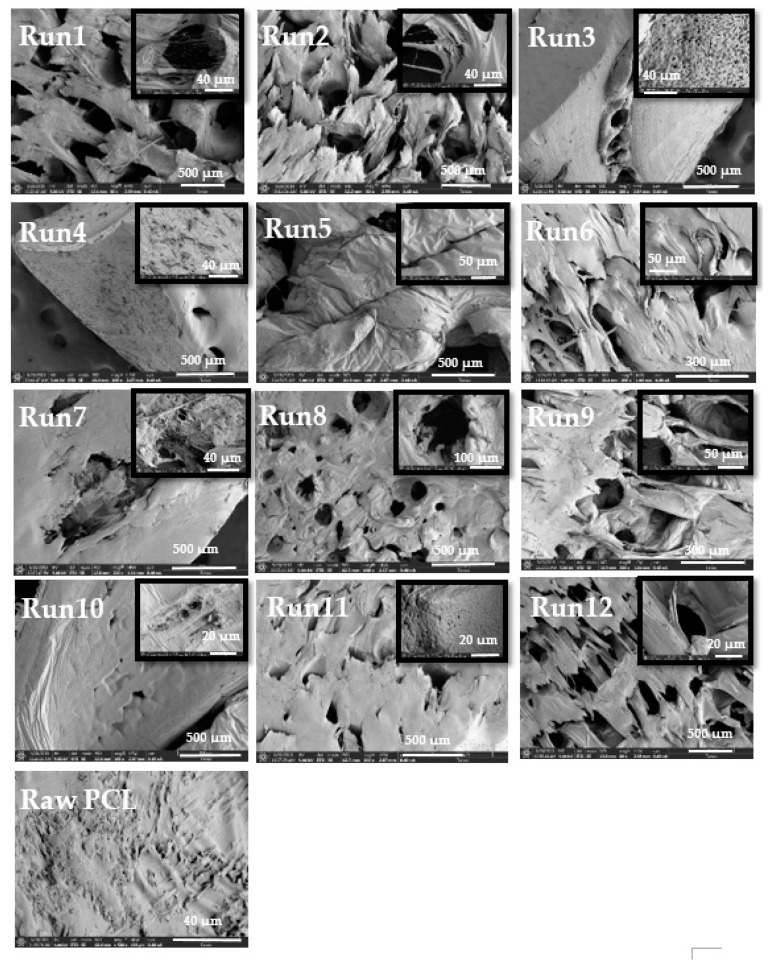
SEM images of composites and raw PCL.

**Figure 4 polymers-11-01390-f004:**
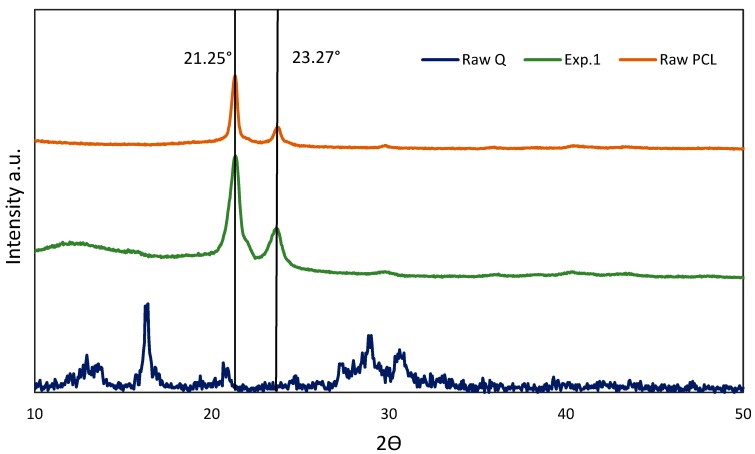
XRD diffraction pattern of composites, raw *Q* and PCL.

**Figure 5 polymers-11-01390-f005:**
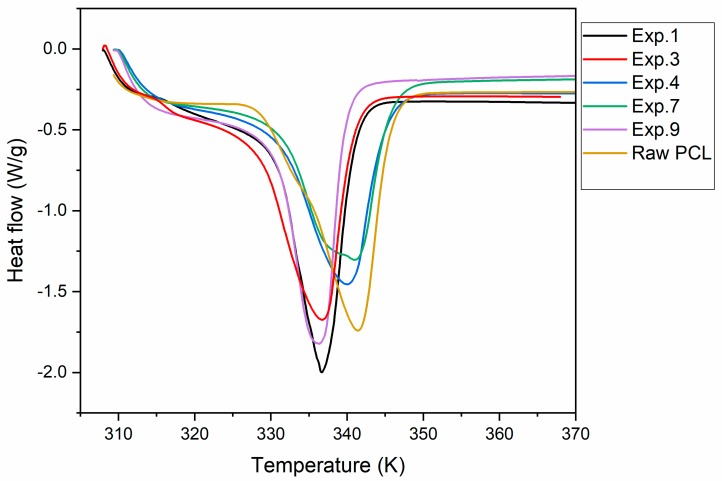
DSC diagrams of composite PCL–quercetin.

**Figure 6 polymers-11-01390-f006:**
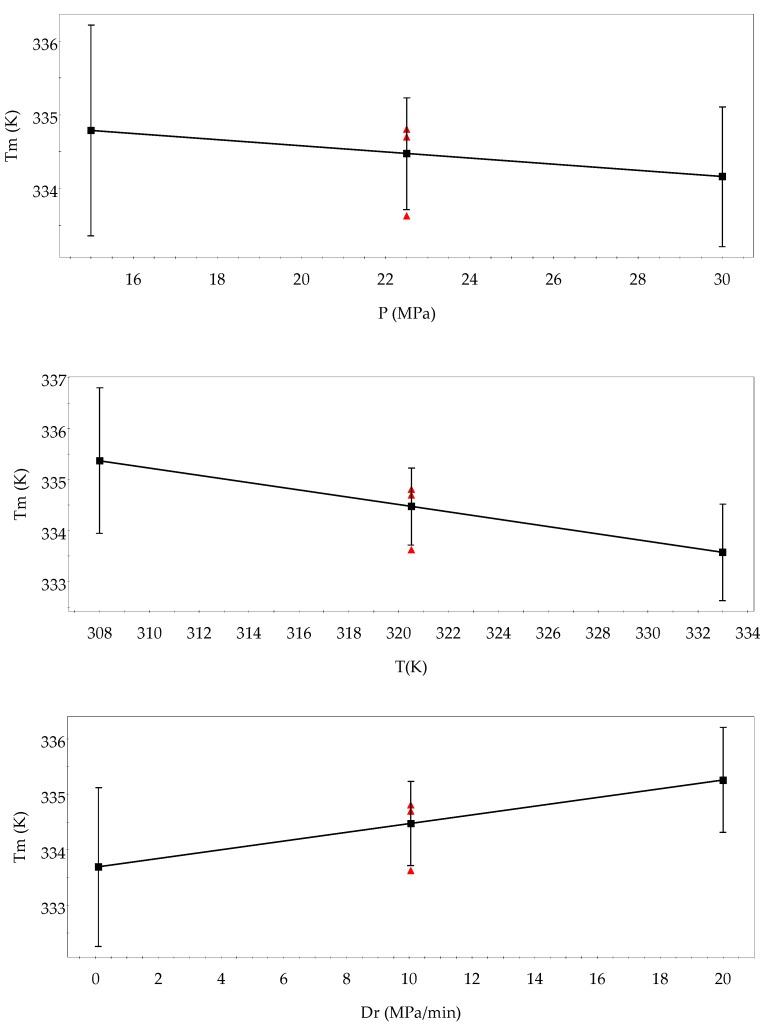
Main effects on *T*_m_ of composite PCL–quercetin.

**Figure 7 polymers-11-01390-f007:**
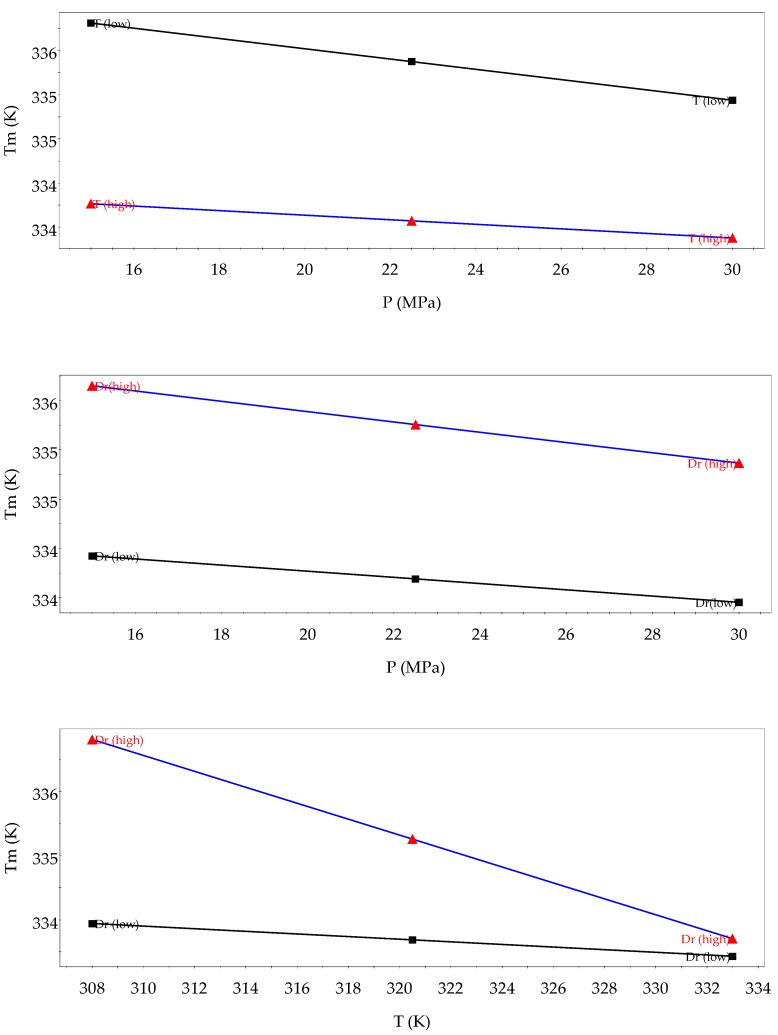
Variable interaction plots on *T*_m_ of composite PCL–quercetin.

**Figure 8 polymers-11-01390-f008:**
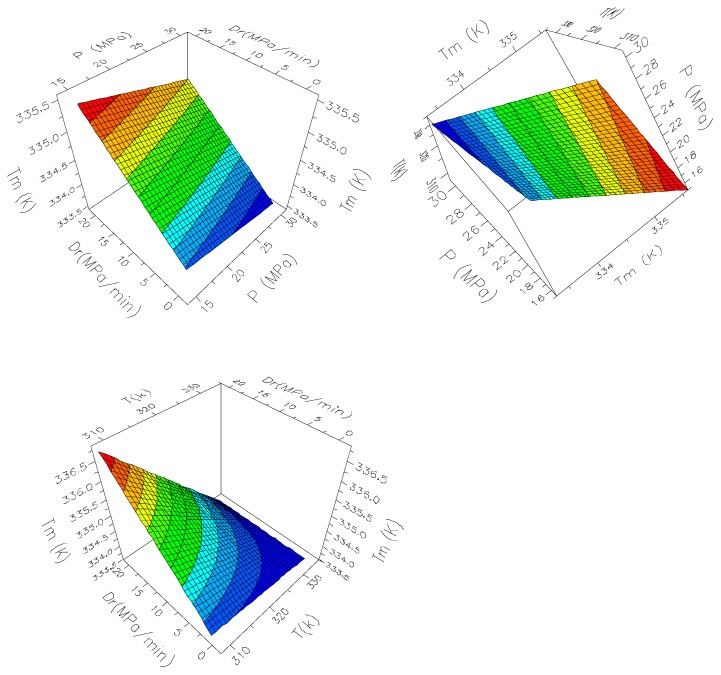
Response surface plots of *T*_m_ of composites.

**Figure 9 polymers-11-01390-f009:**
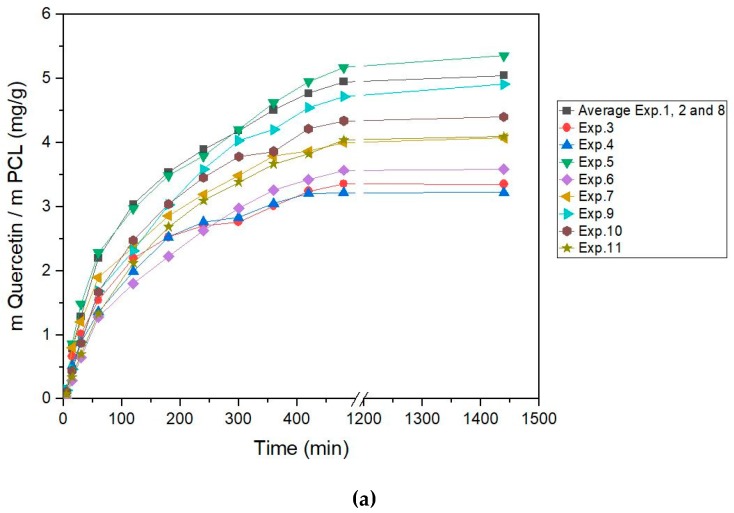
Release tests. (**a**) Release test of *Q* with polymer in experiments 1–11; (**b**) comparison release test raw *Q* and average experiments 1, 2, and 8.

**Figure 10 polymers-11-01390-f010:**
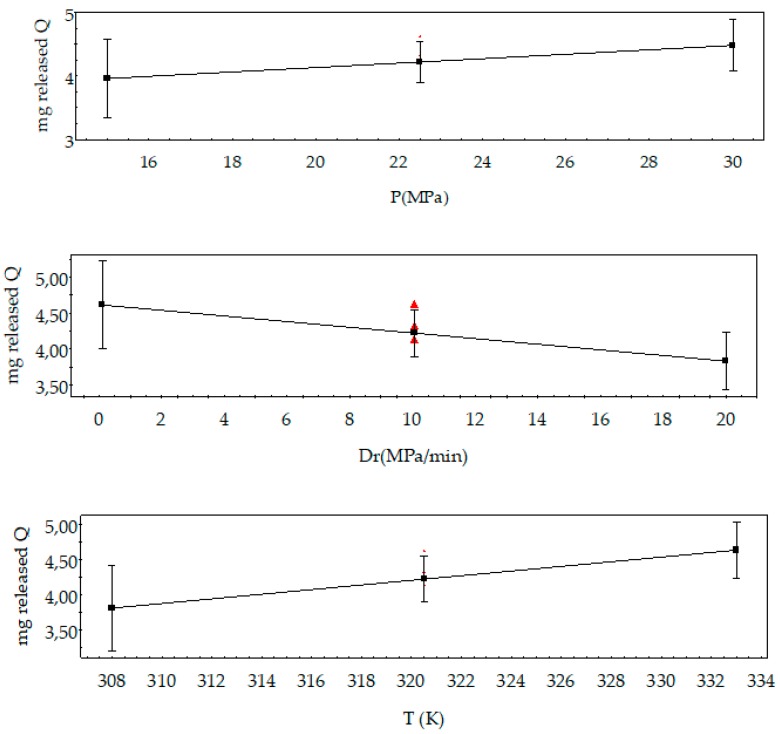
Main effects on released quercetin of the composites.

**Figure 11 polymers-11-01390-f011:**
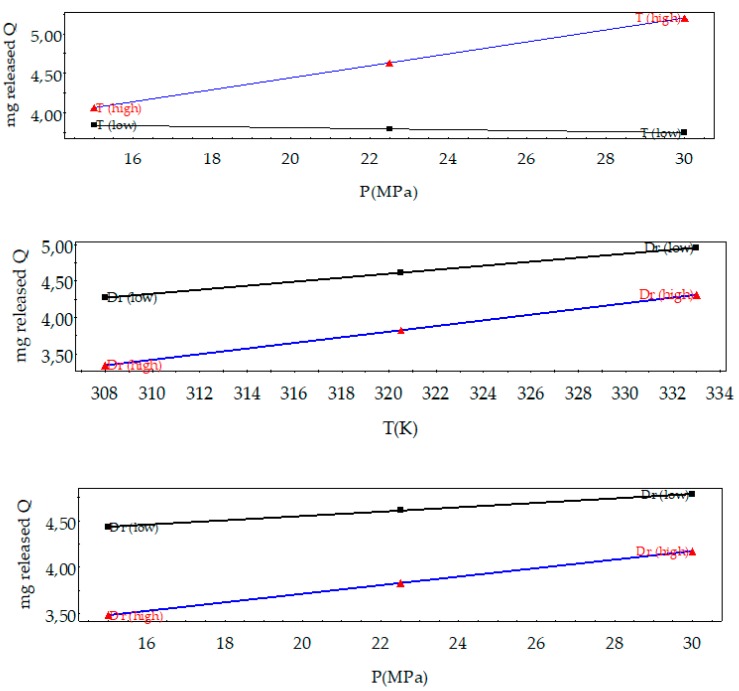
Variable interactions plots on released quercetin of the composites.

**Figure 12 polymers-11-01390-f012:**
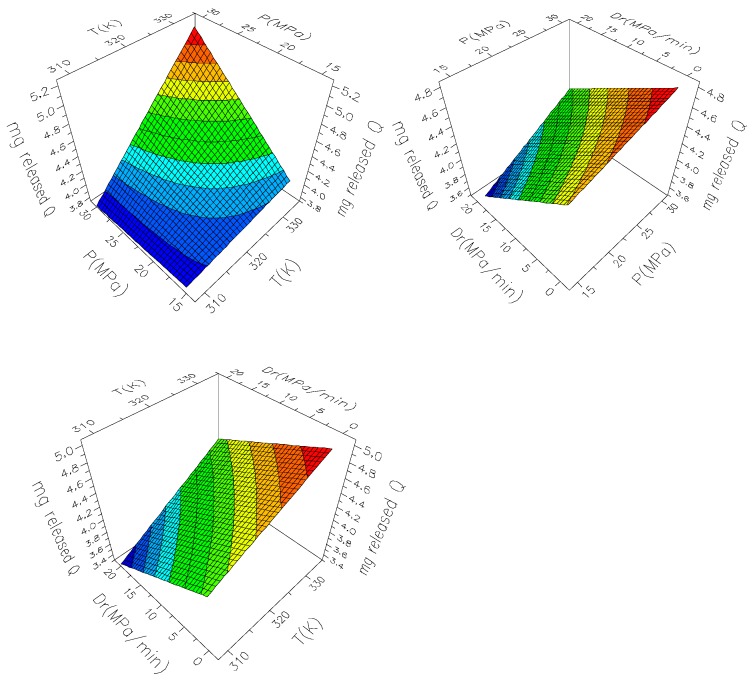
Response surface plots of released quercetin of the composites.

**Table 1 polymers-11-01390-t001:** Two-level assessment for each factor and the calculated effects on responses.

Factors	Low Level	High Level	*T*_m_ Effects	Released *Q* Effects	*H_m_* Effects
*P (*MPa*)*	15	30	−0.44	0.59	−0.61
*T* (K)	308	333	−2.02	0.90	−1.90
*D*r (MPa min^−1^*)*	0.10	20	1.31	−0.75	1.42

**Table 2 polymers-11-01390-t002:** Analysis of variance for the design model of process variables.

Variables		*DF*	*SS*	*MS*	*R* ^2^	*AdjR* ^2^	*Q* ^2^	Significance *F*	Lack of Fit *p*
*T_m_*	Model	9	16.86	1.87	0.95	0.84	0.53	8.72	0.048
	Residual	3	0.91	0.30					
	Error	2	0.85	0.42					
Released *Q*	Model	9	4.17	0.46	0.95	0.84	0.57	9.08	0.049
	Residual	3	0.22	0.07					
	Error	2	0.12	0.06					
*H_m_*	Model	9	379.17	42.13	0.49	−0.52	0.04	0.48	0.79
	Residual	3	191.68	63.90					
	Error	2	88.78	44.40					

**Table 3 polymers-11-01390-t003:** Experimental design and observed responses.

Experiments	*P* (MPa)	*T* (K)	*D_r_* (MPa min^−1^*)*	*T_m_ (*K*)*	mg Released *Q*/g PCL	*H_m_* (J/g)
1	22.5	320.5	2.5	334.70	4.63	85.07
2	22.5	320.5	2.5	333.63	4.33	73.50
3	30	308	20	336.70	3.35	87.39
4	15	308	20	337.10	3.22	83.69
5	30	333	0.1	333.48	5.36	90.58
6	15	333	20	333.91	3.58	87.77
7	15	308	0.1	338.32	3.35	84.57
8	22.5	320.5	2.5	334.81	4.14	85.01
9	30	333	20	333.23	4.91	96.52
10	15	333	0.1	333.61	4.40	80.10
11	30	308	0.1	333.74	4.10	92.69

**Table 4 polymers-11-01390-t004:** Coefficients list of the linear model for *T*_m_ and released quercetin and their significance.

	Coefficient (*T_m_*)	*p* (*T_m_*)	Coefficient (Released *Q*)	*p* (Released *Q*)
Constant	61.40	6.26 × 10^−8^	4.25	2.18 × 10^−5^
*P*	−0.19	0.43	0.26	0.08
*T*	−0.88	0.02	0.39	0.03
*Dr*	0.57	0.04	−0.33	0.04
*P·T*	−5.71 × 10^−3^	0.98	0.23	0.09
*P Dr*	−0.04	0.86	0.06	0.55
*T·Dr*	−0.57	0.06	0.05	0.62

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
