# Peer review of "Foaming of Polycaprolactone and Its Impregnation with Quercetin Using Supercritical CO2"

_polymers, 2019, doi:10.3390/polym11091390_

Round 1

Reviewer 1 Report

Manuscript ID: polymers-563004 - Review Report

In this work “Foaming of polycaprolactone and impregnation with quercetin using 
supercritical CO” the author studies the foaming of PCL using supercritical CO2 and impregnation with quercetin. Though foaming of PCL using supercritical CO2 was studied widely but author has extended the study which is the impregnation part with quercetin. This is a good piece of work but has not organized properly. The work may be accepted if the author can improve it with the below given comments:

1.    The introduction section was not systematic. The author should rearrange explaining what has already been done in foaming of PCL. Then something about quercetin and its past studies. Why it is important to impregnate with quercetin. Literature which was already been published should inserted. Then the author should say what is the novelty of their study in this work.

2.    The resolution of the figures are very bad and it needs a struggle to inderstand the figures.

3.    Some of the figures instead of images the aithor might improve it with a flowchart schematic.

4.    The results especially the morphology, DSC and release part need to be improved.

Author Response

1. The introduction section was not systematic. The author should rearrange explaining what has already      been done in foaming of PCL. Then something about quercetin and its past studies. Why it is important to impregnate with quercetin. Literature which was already been published should inserted. Then the author should say what is the novelty of their study in this work.

As the reviewer suggests the introduction has been rearranged.

2. The resolution of the figures is very bad and it needs a struggle to understand the figures.

As the reviewer suggests the resolution of the figures has been improved.

3. Some of the figures instead of images the author might improve it with a flowchart schematic.

As the reviewer suggests Figure 1 has been substituted by a schematic flowchart

4. The results especially the morphology, DSC and release part need to be improved.

As the reviewer suggests the results has been rewritten. Moreover, melting heat has been introduced as response into the design of experiment.

Reviewer 2 Report

Paper can be re-considered after serious revision in accordance with the following comments:

1. Title should be corrected, for instance “Foaming of polycaprolactone and its impregnation with quercetin using supercritical CO2”.

2. No abbreviation should be used in abstract. Please give abbreviations and their explanation in the main text.

3. line 35 The sentence “..like the swelling, the melting point variation, foaming...[3–5].” Should be corrected, please remove ellipsis before references.

4. Section 2.2. Detailed description of all the obtained samples (including samples 12 and 13) should be given in this section (like first four columns in Table 3).

5. Figures 1, 2 and 4 – quality is unacceptable.

6. Section 2.4.2. What source (radiation) was used in XRD experiments? Please give it.

7. Section 2.4.3. Please give the purity of nitrogen. Please also describe why you use nitrogen instead of inert gas? Are you sure that nitrogen does not react with your materials?

8. Line 164 and below. What is Tm, is it melting point? Please describe it. In this case “m” should be given as subscript, not in the line.

9. Section 3.3.1. Porosity of samples and averages pore size should be obtained quantitatively and presented for all the samples.

10. Figure 3. Please improve the figure, magnification bars are unreadable. Moreover, to compare the structures obtained at different experimental conditions, magnification for all micrographs should be the same, please provide it.

11. Figure 4. Please indicate all the peaks in XRD patterns.

12. Section 3.3.2. Please describe the appearance of the second peak for processed PCL in relation to the unprocessed one.

13. Section 3.3.2. Please give the expanded discussion, why the XRD patterns for impregnated samples does not reveal the presence of Q. See also comment 13.

14. General comment. What was the ratio between PCL and Q in your experiments? Please obtain and give it for all the samples.

15. Sections 2.4.3 and 3.3.3. How the Tm value was calculated from the DSC curves, please describe in detail.

16. Sections 2.4.3 and 3.3.3. The value of melting heat should be also obtained using DSC curves and duly discussed.

17. Figures 6 and 11 are hard for understanding. Please give the data in more appropriate form.

Author Response

Paper can be re-considered after serious revision in accordance with the following comments:

1. Title should be corrected, for instance “Foaming of polycaprolactone and its impregnation with quercetin using supercritical CO2”

The title has been corrected.

2. No abbreviation should be used in abstract. Please give abbreviations and their explanation in the main text.

As the reviewer suggest abbreviations have been removed from the abstract.

3. line 35 The sentence “...like the swelling, the melting point variation, foaming...[3–5].” Should be corrected, please remove ellipsis before references.

As the reviewer suggests the ellipsis has been removed.

4. Section 2.2. Detailed description of all the obtained samples (including samples 12 and 13) should be given in this section (like first four columns in Table 3).

Samples 12 and 13 correspond to PCL processed with CO2 and raw PCL. For this reason, were not included in Table 3. Anyway the names of experiments have been unified.

5. Figures 1, 2 and 4 – quality is unacceptable.

Quality of figures have been improved.

6. Section 2.4.2. What source (radiation) was used in XRD experiments? Please give it.

As the reviewer suggests the source of radiation has been indicated.

7. Section 2.4.3. Please give the purity of nitrogen. Please also describe why you use nitrogen instead of inert gas? Are you sure that nitrogen does not react with your materials?

The analyses were made by the Titania enterprise according to standard ISO 11357-3 [2018] where N2 that is an inert gas is used. The purity of the gas is 99.999%. This fact was included into the manuscript.

8. Line 164 and below. What is Tm, is it melting point? Please describe it. In this case “m” should be given as subscript, not in the line.

Effectively Tm is melting point. As the reviewer suggests “m” has been changed to subscript.

9. Section 3.3.1. Porosity of samples and averages pore size should be obtained quantitatively and presented for all the samples.

In this paper porosity has been considered as qualitative way due to it is a preliminary study of PCL foaming. In later investigation, where a determined scaffold is wanted to build, an analyses in depth of porosity and pore size distribution will be made by adsorption of N2.

10. Figure 3. Please improve the figure, magnification bars are unreadable. Moreover, to compare the structures obtained at different experimental conditions, magnification for all micrographs should be the same, please provide it.

As the reviewer suggests the figures have been improved and as possible as we could, with the same magnification.

11. Figure 4. Please indicate all the peaks in XRD patterns.

As the reviewer suggests the peaks of PCL were included and the numerous peaks of Q were included in the manuscript.

12. Section 3.3.2. Please describe the appearance of the second peak for processed PCL in relation to the unprocessed one.

The second peak exists also in unprocessed PCL but has lower intensity.

13. Section 3.3.2. Please give the expanded discussion, why the XRD patterns for impregnated samples does not reveal the presence of Q. See also comment 13.

In the manuscript authors discuss the possibilities of the absence of Q peaks in the pattern. On one hand, it could be due to Q is changed to amorphous nature although authors seem that is unlikely, and on the other and the proportion of polymer is quite higher than Q so peaks of Q were not observed.

14. General comment. What was the ratio between PCL and Q in your experiments? Please obtain and give it for all the samples.

In the manuscript amount of PCL and Q are included although not in the ratio format. The ratio is 50:1 PCL: Q. It has been included in the manuscript.

15. Sections 2.4.3 and 3.3.3. How the Tm value was calculated from the DSC curves, please describe in detail.

Using DSC it is possible to observe fusion and crystallization events as well as glass transition temperatures so Tm is proportioned by DSC analyses. These analyses were carried out by Titania enterprise and a list of Tm is given. DSC can also be used to study oxidation, as well as other chemical reactions.

16. Sections 2.4.3 and 3.3.3. The value of melting heat should be also obtained using DSC curves and duly discussed.

As the reviewer suggests the melting heat was included in the manuscript and was discussed.

17.  Figures 6 and 11 are hard for understanding. Please give the data in more appropriate form.

Interaction plots are needed to observe if the parameter are independent or not. The graphics is a way to see this fact in an easy way. If the lines have the same slope, no interaction between variables is produced and if the lines are crossed each other there is interaction. Anyway the figures are explained in the manuscript.

Round 2

Reviewer 2 Report

Comments 1 - 4, 7 - 14 were answered well.

Comment 5 - Figures quality is still poor.

Comment 6 - Please give the source in correct form (CoKα).

Comment 15. Calculation of Tm from DSC curves should be described in detail in manuscript.

Comment 16. Melting heat and its relation with the sample stucture and preparation route should be discussed in text.

Comment 17. I still ask authors to give these furures in more appropriate form.

Author Response

Reviewer #2:

Comments 1-4,7-14 were answered well.

Comment 5-Figures quality is still poor.

Figure 4 was improved. Authors think the rest of figures are enough quality.

Comment 6-Please give the source in correct form (CoKa).

As the reviewer suggests the source was introduced in correct form.

Comment 15.Calculation of Tm from DSC curves should be described in detail in manuscript.

DSC analysis measure the heat flow produced in a sample when it is heated. Thus melting temperature peak is proportioned. Authors consider that there is no the objective of the work to describe the basis of DSC analysis that is a tool widely used by researchers in polymer investigation.

Comment 16. Melting heat and its relation with sample structure and preparation route should be discussed in text.

Once melting heat response was not significant, authors consider that no relationship could be established with respect to structure and preparation route (P, T and Dr) and melting heat. For that reason melting heat was included in the design but was removed from discussion.

Comment 17.I still ask author to give these furures in more appropriate form.

Authors consider that the interaction plots are an easy way to observe if the parameters are independent or not. This kind of graphic is recommended in design of experiments. Moreover is complementary information so authors think that it should be included in this form.

Round 3

Reviewer 2 Report

Authors nearly answered my comments. Paper may be accepted in present form.